# Late Paleozoic oxygenation of marine environments supported by dolomite U-Pb dating

Michal Ben-Israel [1,2], Robert M. Holder [3], Lyle L. Nelson [4,5],
Emily F. Smith [6], Andrew R. C. Kylander-Clark[7] & Uri Ryb [1]✉

Understanding causal relationships between evolution and ocean oxygenation hinges on reliable reconstructions of marine oxygen levels, typically from redox-sensitive geochemical proxies. Here, we develop a proxy, using dolomite U–Pb geochronology, to reconstruct seawater U/Pb ratios. Dolomite samples consistently give U–Pb dates and initial $^{207}Pb/^{206}Pb$ ratios lower than expected from their stratigraphic ages. These observations are explained by resetting of the U–Pb system long after deposition; the magnitude of deviations from expected initial $^{207}Pb/^{206}Pb$ are a function of the redox-sensitive U/Pb ratios during deposition. Reconstructed initial U/Pb ratios increased notably in the late-Paleozoic, reflecting an increase in oxygenation of marine environments at that time. This timeline is consistent with documented shifts in some other redox proxies and supports evolution-driven mechanisms for the oxygenation of late-Paleozoic marine environments, as well as suggestions that early animals thrived in oceans that on long time scales were oxygen-limited compared to today.

The oxygenation of Earth's atmosphere and oceans and the emergence and evolution of life are pivotal episodes in Earth history[1–5]. Two step-wise increases in atmospheric oxygen level at ~2.4-2 and 0.8-0.55 Ga (billions of years) are generally agreed upon and supported by an abundance of geochemical and sedimentological evidence[2]. However, the timing in which different marine habitats became oxygenated, and therefore the causal relationship between such oxygenation and evolutionary steps are intensely debated[6,7]. Hypothesized timelines of marine oxygenation range from a step-wise increase in $O_2$ between the Neoproterozoic and Cambrian, parallel to the radiation of animal life[4,8–11] (possibly reaching values as high as modern oceans by the early Cambrian[8]); to minor (0.5–10% of the modern ocean levels[12–14]), local, and/or episodic oxygenation[15–20], prior to the late Paleozoic-early Mesozoic eras, when evolutionary steps such as the expansion of vascular plants or the proliferation of phytoplankton drove major global redox changes[17–23]. To test these hypotheses, reconstructions of the timeline and extent of oxidation of marine environments from redox sensitive geochemical proxies are imperative.

Interpretations of concentrations and isotope ratios of redox-sensitive elements measured in sedimentary rocks as primary signals, reflecting the conditions in ancient depositional environments, have been criticized for the possibility of post-depositional signal alteration[24–26]. This problem is accentuated in carbonate rocks and minerals, which are notorious for element and isotope exchange in diagenetic and metamorphic environments[22,27]. Dolomite, a common alteration product of Ca-carbonate minerals through interaction with seawater in diagenetic environments, is often seen as an unlikely candidate to preserve primary redox-sensitive signals[27]. Nevertheless,

[1]The The Fredy & Nadine Herrmann Institute of Earth Sciences, The Hebrew University of Jerusalem, Jerusalem, Israel. [2]Department of Life and Environmental Sciences, University of California, Merced, CA, USA. [3]Department of Earth and Environmental Sciences, University of Michigan, Ann Arbor, MI, USA. [4]Department of Earth Sciences, Carleton University, Ottawa, Ontario, ON, Canada. [5]Department of Earth, Atmospheric, and Planetary Sciences, Massachusetts Institute of Technology, Cambridge, Massachusetts, MA, USA. [6]Department of Earth and Planetary Sciences, Johns Hopkins University, Baltimore, MD, USA. [7]Department of Earth Science, University of California, Santa Barbara, California, CA, USA. ✉e-mail: uri.ryb@mail.huji.ac.il

here we demonstrate that such information can be carried beyond the 'veil' of post-depositional alteration by the U and Pb isotope compositions of dolomites. We examine new and previously published U and Pb isotope analyses of dolomites from Mesoproterozoic to Cretaceous marine strata, and show that (1) dolomite U–Pb dates are commonly reset during burial diagenesis 10–100's of Ma (millions of years) after deposition; (2) in this process, Pb isotope values are homogenized in a closed-system environment; and (3) calculated initial Pb isotope ratios can be used to infer the redox-sensitive $^{238}U/^{206}Pb$ ratio in the dolomite-precursor carbonate mineral phase, recorded in marine depositional and early diagenetic environments.

Marine carbonate minerals are desirable targets for U–Pb dating because they are prevalent authigenic phases throughout the geological record[28]. U–Pb dating is based on the decay of U parent nuclides, $^{238}U$ and $^{235}U$, to their ultimate daughter isotopes, $^{206}Pb$ and $^{207}Pb$, respectively. In an ideal scenario, the initial Pb-isotope composition of a sample is homogeneous whereas the U/Pb ratio is spatially heterogenous. Analyses from a sample will plot along a negative linear trend in $^{207}Pb/^{206}Pb$ vs. $^{238}U/^{206}Pb$ composition space (also known as the 'Tera-Wasserburg' or 'inverse' concordia diagram)[29]. The intercept of this trend on the $^{207}Pb/^{206}Pb$ axis ($^{238}U/^{206}Pb = 0$, therefore [U] = 0) defines the whole-rock $^{207}Pb/^{206}Pb$ ratio at the time of (re)crystallization ("initial $^{207}Pb/^{206}Pb$"). The lower intercept between this trend and the concordia curve will indicate the U–Pb date of the sample, interpreted as the time that passed since the closure of the system[29,30].

## Results and discussion

We analyzed Paleozoic and Meso- to Neoproterozoic dolomite samples from the Colorado Plateau and the Death Valley regions, determining their U–Pb dates and initial $^{207}Pb/^{206}Pb$ using the Tera-Wasserburg concordia scheme (see Methods for detailed description). Using in-situ laser ablation inductively coupled plasma mass spectrometry (LA-ICP-MS), U and Pb isotope measurements were made on a variety of replacive dolomite mineral fabrics associated with depositional and early diagenetic environments (fabric-retentive) or possibly with late diagenetic environments (fabric-destructive), while avoiding void-filling epigenetic dolomite (e.g., veins). We combined these results with U–Pb data from similar dolomite fabrics in the Rocky Mountains[31], Eastern Levant[32], Arabian Platform[33] and Northwest Sichuan Basin[34] to create a more globally representative dataset. These data represent six different basins and span a period of ~1.2 billion years, from the Mesoproterozoic Eon to the Late Cretaceous Epoch (Supplementary Dataset S1).

Carbonate minerals forming in marine depositional and shallow diagenetic environments are expected to inherit the initial $^{207}Pb/^{206}Pb$ ratios of seawater. In turn, the $^{207}Pb/^{206}Pb$ value of pre-industrial seawater is expected to follow the composition of terrestrial Pb as it evolved over time[35], with minor scatter around the average value reflecting the heterogeneity in terrestrial Pb sources and the limited mixing of Pb in the ocean. We estimate the scatter around $^{207}Pb/^{206}Pb$ ratios of terrestrial Pb to be <0.02, based on the observed range of $^{207}Pb/^{206}Pb$ values in modern deep ocean water masses, which preserve pre-industrial Pb isotope values[36–42]. If the U–Pb system in dolomites closed during or shortly after deposition, the U–Pb date would correspond to the depositional age of the sediment, and the initial $^{207}Pb/^{206}Pb$ would plot within the uncertainty-envelope of the terrestrial Pb evolution trend. However, with few exceptions, calculated U–Pb dates are significantly younger (by $10^{7–8}$ years, Fig. 1a) than the depositional ages of the sedimentary strata, and calculated initial $^{207}Pb/^{206}Pb$ ratios of dolomites plot well below the terrestrial and seawater Pb evolution trend (Fig. 1b).

Younger than expected U–Pb dates and lower than expected $^{207}Pb/^{206}Pb$ initial values in dolomite samples indicate a significant modification to the U–Pb system long after deposition of the original carbonate strata, possibly associated with dolomitization or dolomite-

recrystallization during late-stage burial diagenesis. Alteration in deep burial environments through open-system isotope exchange with seawater has been suggested to modify the carbonate oxygen and clumped isotope compositions of massive dolomites in general, and specifically, the set of Paleozoic dolomites from the Colorado Plateau analyzed here[43]. However, if during such modification the Pb isotope system was similarly open to exchange with seawater, then initial $^{207}Pb/^{206}Pb$ values would have been reset back to terrestrial values corresponding to the time of modification, contrary to our observation (Fig. 1b). Other external sources for Pb, such as river waters or deep-seated groundwater are also unlikely, as $^{207}Pb/^{206}Pb$ ratios in these waters are also expected to plot close to terrestrial values[44,45]. The most likely source for the lower initial $^{207}Pb/^{206}Pb$ is the dolomite-precursor rocks themselves, which have higher U/Pb ratios than average crust and therefore their bulk $^{207}Pb/^{206}Pb$ values diverge from the terrestrial trend as U decays (Fig. 2a, b). During the late-stage diagenetic modification, Pb was redistributed among dolomite minerals and isotopically homogenized in a system effectively closed to external Pb isotopic exchange, resulting in a new, lower "initial" $^{207}Pb/^{206}Pb$ value (Fig. 2b). In this process, U concentrations could also have been re-distributed and partly homogenized in either open or closed system conditions, but some U/Pb heterogeneity was maintained allowing U-Pb discordia to be defined. Following this diagenetic Pb homogenization, further U decay in a closed system ultimately resulted in a younger than depositional U-Pb date, reflecting the time of modification (Fig. 2c). Supporting this interpretation of dolomite U-Pb systematics, a recent study demonstrated that the isotopic composition of U in dolomites is acquired by equilibration with the final diagenetic fluid at the time of alteration, or soon after, and has evolved in a closed system since[46]. The extent to which the bulk $^{207}Pb/^{206}Pb$ ratios were lowered between the time of deposition and latest stage of diagenesis depends on the bulk U/Pb ratio of the dolomite-precursor rock, which in turn depends on the redox state of the environment in which it precipitated[47]. Larger bulk U/Pb ratios will result in larger deviations from the average marine $^{207}Pb/^{206}Pb$ ratio as U decays.

Dolomite samples in our data compilation include fabric-retentive dolomites that have been interpreted as products of early dolomitization[48,49]. Yet, many of these samples apparently underwent a closed-system homogenization of Pb isotope values, $10^{7–8}$ years after deposition (Fig. 1A). This seeming contradiction can be explained if the resetting of the U-Pb system was associated with recrystallization of (proto)dolomite rather than initial dolomitization, at least for the fabric-retentive dolomites in our compilation. Recrystallization of fabric-retentive and -destructive dolomites would be in line with studies included in our dataset, which indicate open-system isotope exchange of carbonate oxygen, and re-setting of clumped isotope compositions in deep burial environments[33,43]. Therefore, we hypothesize that during late-stage diagenesis there was closed-system homogenization of Pb, while oxygen and clumped isotope compositions were altered under open-system behavior, as oxygen is a highly abundant and mobile element compared to Pb, and therefore more easily buffered by diagenetic fluids. Consistent with this explanation, late recrystallization has been recently suggested to reset the U–Pb dates, and oxygen and clumped isotope compositions within a set of fabric-retentive dolomites[33]. Alternatively, we cannot rule out the possibility that oxygen and Pb isotope systems have responded to two separate late-stage alteration events. For example, oxygen and clumped isotopes were altered in dolomite samples during contact metamorphism around the Alta Stock (Utah, USA), while the U–Pb dates were not[31].

The calculated initial $^{207}Pb/^{206}Pb$ values in dolomite samples reflect the homogenized bulk-average $^{207}Pb/^{206}Pb$ evolved in the dolomite-precursor rock at the time of closed-system alteration. This value is expected to be driven further below the terrestrial/marine

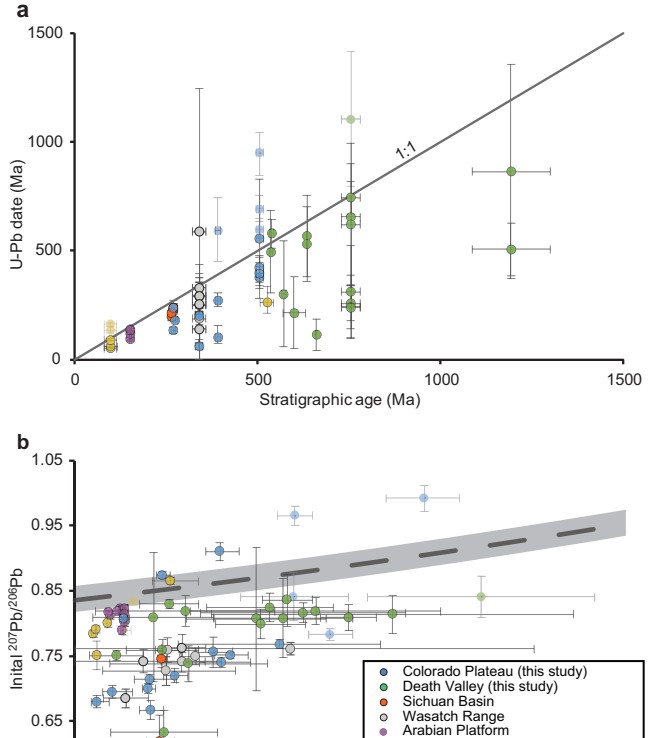

**Fig. 1 | Calculated U–Pb dates and initial $^{207}Pb/^{206}Pb$ ratios. a** Comparison between calculated U–Pb dates and stratigraphic ages of the carbonate strata. Most calculated values plot below the 1:1 ratio line indicating that they are significantly younger compared to stratigraphic ages. Otherwise, semi-transparent points mark samples with U–Pb dates significantly older than their stratigraphic ages, possibly due to contribution of Pb from a detrital source. Large uncertainties in U-Pb dates, presented as 2 Standard Deviation (SD) error bars, are common to carbonate samples that typically suffer from low ranges of $^{238}U/^{206}Pb$. **b** Calculated initial $^{207}Pb/^{206}Pb$ ratios over time. Most values plot below the predicted line of isotopic evolution of terrestrial Pb[35], consistent with Pb-isotope re-homogenization 10–100's of millions of years after deposition. Four samples plot above the trend, of which two have U–Pb dates older than their stratigraphic ages (semi-transparent). Gray rectangle is an uncertainty envelope for marine $^{207}Pb/^{206}Pb$ values, considering the observed range of values among pre-industrial water sources[38,40–42,45]. Error bars are 2 SD. Source data are provided as a Source Data file.

initial $^{207}Pb/^{206}Pb$ trend with a longer interval between deposition and alteration time, and higher bulk-average initial $^{238}U/^{206}Pb$ in the dolomite precursor rock. Considering that the time gap between original deposition and alteration is known from the difference between stratigraphic ages and U–Pb dates, we can estimate the initial $^{238}U/^{206}Pb$ in the dolomite precursor. To do that, we assume that the precursor mineral's initial $^{207}Pb/^{206}Pb$ ratio was the terrestrial $^{207}Pb/^{206}Pb$ ratio during the deposition of the dolomite-precursor carbonate minerals[35]. We then calculate the corresponding initial bulk-averaged $^{238}U/^{206}Pb$ ratio that is required to drive $^{207}Pb/^{206}Pb$ from that value to the observed initial value in the dolomite after a time interval defined by the difference between stratigraphic ages and U–Pb dates. We repeat this calculation for all samples in which the initial $^{207}Pb/^{206}Pb$ ratio is lower and the U–Pb date is younger than the expected marine value (54 out of 65 dolomite fabrics examined), and consider the reconstructed values to approximate bulk averaged $^{238}U/^{206}Pb$ ratios in the dolomite-precursor rock.

Post depositional alteration of geochemical signals is common in ancient rock records. The approach we develop here, to reconstruct

redox sensitive $^{238}U/^{206}Pb$ ratio in depositional and early diagenetic environments from the redox insensitive initial $^{207}Pb/^{206}Pb$ ratio, is likely applicable to other authigenic mineral records (e.g., calcite, apatite, and iron oxides). In fact, 'younger-than-depositional' U-Pb dates and 'lower-than-expected' initial $^{207}Pb/^{206}Pb$ values have been recently reported in bio-apatite[50].

Reconstructed $^{238}U/^{206}Pb$ values remain below 14 during the Proterozoic Eon and early Paleozoic Era, and show a distinct increase in range, reaching maximum values of ~25-376 after ~400 Ma, following the reconstructed trajectory of atmospheric $pO_2$ (Figs. 3a, 4). One possible explanation for intervals with elevated $^{238}U/^{206}Pb$ is that these periods have been dominated by aragonite precursor for dolomite, in which U is more compatible than in calcite[51]. However, reconstructions of the primary carbonate minerals deposited in shallow marine environments suggest that alterations between dominantly aragonite and calcite precipitates during the Neoproterozoic and Phanerozoic eons do not correlate to the step variation in the range of reconstructed $^{238}U/^{206}Pb$ shown here[52,53], thus contradicting this explanation as the major driver of the observed variability.

An alternative explanation for the increase in the range of reconstructed $^{238}U/^{206}Pb$ is that the availability of dissolved U and Pb species in carbonate crystallization environments has changed dramatically. It is not likely that such a change is the result of variation in the supply of U and Pb to the ocean, as oxidative continental weathering of these elements has been dominant since the Paleoproterozoic[1,54]. Instead, this change could have resulted from rising levels of oxygen in the water column and pore fluids of the upper sediment pile, which would increase the supply of dissolved $U^{6+}$ and decrease that of Pb (now scavenged by Fe-Oxyhydroxides e.g.[55]) to the crystallization site and thereby drive the $^{238}U/^{206}Pb$ of dolomite-precursor carbonate minerals higher[47]. Supporting this hypothesis, lower concentrations of U have been recorded in carbonates during intervals of expanded oceanic anoxia within the Phanerozoic[56,57].

The above interpretation is based on the observed initial $^{207}Pb/^{206}Pb$ ratio, which is relatively impervious to alteration after dolomite (re)crystallization, and specifically, to false-positive oxygenation signals (higher $^{238}U/^{206}Pb$) recorded as samples were approaching the surface during exhumation. However, it may still record signals less-oxic (lower $^{238}U/^{206}Pb$) than surface environments, if the dolomite precursor minerals formed in anoxic pore fluids beneath the sediment-water interface. Much of the observed scatter in reconstructed $^{238}U/^{206}Pb$ values among samples from the same rock units can be explained as reflecting variability in pore-water redox conditions. Such could also be the case for dolomites associated with the degradation of microbial mats in the shallow sediment and/or with local anoxia e.g., 33,49. Nevertheless, the first order <400 Ma oxygenation signal is larger than this scatter (Figs. 3a, 4). An example of this might be seen in the Jurassic microbial-associated dolomites which show a moderately lower range of reconstructed $^{238}U/^{206}Pb$ values, compared to slightly older and younger fabric-destructive (massive) dolomites: the values from these microbial-associated dolomites are still higher than the ranges observed in >400 Ma samples (Fig. 3a), suggesting that a signal of elevated U/Pb$_{seawater}$ might still be recorded in such environments. Other non-microbial-associated dolomite textures, record significantly higher values of reconstructed $^{238}U/^{206}Pb$ that are expected under more oxygenated depositional or early diagenetic environments only after ~400 Ma (Figs. 3a, 4). Notably, the variation in reconstructed $^{238}U/^{206}Pb$ values postdates the Neoproterozoic-Cambrian transition between fabric-retentive and -destructive dolomite textures, hypothesized to have resulted from the reduction of the extent of microbial mats capping the sediment[48,49]. However, given that microbial mats played a persisting role in the carbonate factory throughout early Paleozoic marine depositional environments e.g.58, it is possible that in some settings microbial mats continued to modulate $^{238}U/^{206}Pb$ of carbonate precipitates during this interval.

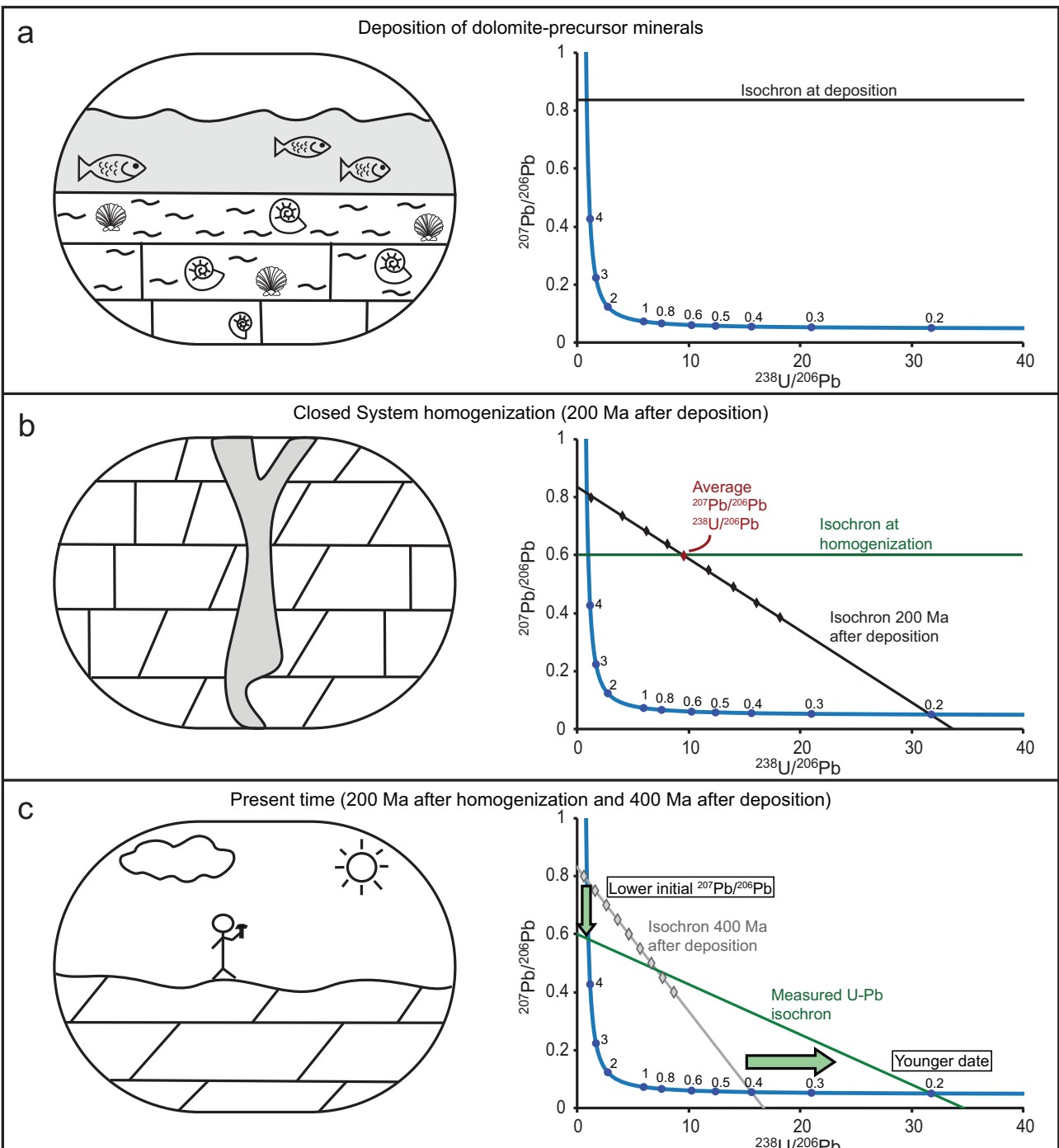

**Fig. 2 | Proposed stages in the evolution of U–Pb system in dolomite precursor carbonate minerals and dolomite, deposited at 400 Ma and altered at 200 Ma, respectively.** Changes to the U–Pb system over time are presented using Tera-Wasserburg concordia diagrams. In each plot, the blue line is the concordia curve with blue dots representing time in Ga. **a** During deposition of carbonate minerals, the initial $^{207}Pb/^{206}Pb$ ratio is inherited from seawater. **b** In burial diagenetic environments, long after deposition and following some isotopic decay within the U–Pb system, Pb is redistributed and isotopically homogenized within the sample in a closed system, and dolomite minerals acquire the system-average $^{207}Pb/^{206}Pb$ ratio. **c** Following late-stage diagenetic Pb homogenization, isotopic decay within the dolomite proceeds in a closed system to the present, resulting in a U–Pb date younger than the depositional age and an initial $^{207}Pb/^{206}Pb$ ratio lower than expected initial seawater composition. Gray rhombs and line represent the isochron that would result if there was no alteration.

Accepting the hypothesis that the higher reconstructed $^{238}U/^{206}Pb$ ratios represent an increase in marine dissolved $O_2$, the question remains: Which parts of the ocean were oxygenated? Opposite redox sensitivities of dissolved U and Pb means that the U/Pb ratio may be affected by both global trends (e.g., the extent of oceanic anoxia) or more local redox conditions (e.g., shallow marine and pore water environments). These possibilities are not mutually exclusive, and

both are supported by independent datasets (Fig. 3b–d). After ~420 Ma, increased oxygenation of deep marine basins is supported by the increase in $Fe^{3+}/\Sigma Fe$ in hydrothermally-altered submarine basalts[23]. At approximately the same time, oxygenation of shallow marine and early diagenetic environments is supported by decreasing Ce/Ce* and increasing I/Ca ratios in marine carbonate samples[17,22], as well as by the observation of limited bioturbation extending at least 120 Ma into the

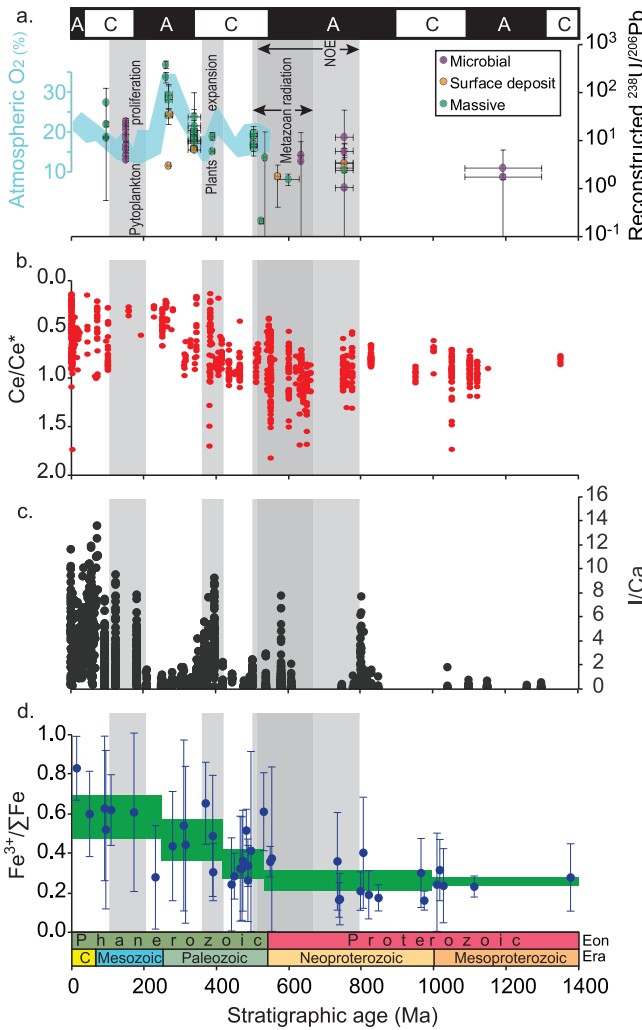

**Fig. 3 | Selected records supporting <400 Ma oxygenation of marine environments.** **a** Reconstructed $^{238}U/^{206}Pb$ ratios in dolomite-precursor minerals, color-coded by texture. Upper bar presents the dominant Ca-carbonate precipitant in marine environments where 'A' is aragonite and 'C' is calcite[53,70]. Cyan trend is reconstructed atmospheric $pO_2$ based on compilation of model results in[71]. Error bars are 2 Standard Error (SE). Source data are provided as a Source Data file. **b** Ce anomaly values measured in marine carbonates[22]. **c** I/Ca ratios measured in marine carbonates[17]. **d** $Fe^{3+}/\sum Fe$ ratios measured in hydrothermally altered basalts (green boxes are 2SE uncertainty envelopes)[23]. Semi-transparent gray bars indicate the timing of the hypothesized Neoproterozoic Oxygenation Event (NOE)[2], Neoproterozoic-Cambrian radiation of metazoan life[6], vascular plants expansion[72], and phytoplankton proliferation[73].

Phanerozoic eon[59]. Thus, reconstructed $^{238}U/^{206}Pb$ values are broadly consistent with data sets that show increases in oxygenation within both water and/or sediment in shallow and deep ocean environments in the late Paleozoic.

Proxy-based reconstructions of the oxygenation of marine environments typically follow one of three patterns leading to persistent oxygenation of the modern ocean: (1) A step increase during the NOE (0.8–0.55 Ga)[e.g. 6,11,60]; (2) Episodic oxygenation between the NOE and Cambrian[e.g. 15,16]; and (3) gradual or episodic increase in oxygenation during the mid–late Paleozoic[e.g. 17,21–23]. These differences may reflect the complex structure of the early Paleozoic oxycline, in which shallow remineralization of organic material reduced continental shelves and left deeper basins relatively oxidized[17], or possibly different sensitivities of elements and the records that contain them to post-depositional alteration[25]. In this context, the fact that reconstructed $^{238}U/^{206}Pb$ ratios, which are resistant to the effects of 'false positive

oxygenation', do not appear to record oxygenation signals during the Neoproterozoic and Cambrian within multiple samples from different formations and different paleoenvironments, at a first order, supports the model that persistent oxygenation of shallow and deep marine environments was delayed until the Devonian (Fig. 3). This scenario does not necessarily preclude the possibilities of minor (<10% or the modern oxic ocean), local, and/or episodic (1–10 Ma) oxygenation events during the NOE, which did not register in the compiled $^{238}U/^{206}Pb$ and Ce/Ce* records shown here, but could have had critical ecological impacts vital for the Cambrian radiation[12].

Several independent datasets now support the oxygenation of marine environments at or after ~400 Ma (Fig. 3). Possible driving mechanisms for this process include: (1) The Devonian expansion of vascular plants, ultimately bringing atmospheric $pO_2$ to peak value towards the end of the Paleozoic[18,21,61]; and (2) early Mesozoic increase in the size of eukaryote plankton, leading to more efficient export of organic carbon to the deep ocean, and thus removing reducing power from the upper water column[17,62]. Regardless of the specific mechanism, the persistent oxygenation of marine habitats long after the Neoproterozoic Era is consistent with changes in organic carbon burial efficiency resulting from novel evolutionary steps. It is also consistent with previous suggestions that several major evolutionary and ecological steps in the diversification of animal life took place within a largely oxygen-limited ocean, in which $O_2$ concentrations were sufficient to overcome functional and ecological thresholds, but insufficient to register in a series of redox proxies, possibly since they were minor (<10%) or existed temporary and/or locally relative to the modern oxic ocean[7,14,20].

## Methods

We collected 15 dolomite rock samples from localities around the Death Valley region. In addition, we used 14 dolomite rock samples from the Paleozoic sedimentary section from the Colorado Plateau, originally collected by Ryb and Eiler[43]. Sample details are available in Supplementary Dataset S2. U–Pb analyses were performed on specific dolomite fabrics (e.g., matrix and various allochems) on 100 μm thick thin-sections. Eighteen dolomite fabrics in samples from the Colorado Plateau samples were analyzed at UCSB using a Nu Plasma 3D multi-collector ICP-MS coupled with a Photon Machines Analyte 193 nm ATLEX-SI 193 nm ArF excimer laser ablation system with a double volume HelEx II cell. Seventeen dolomite fabrics in samples from the Death Valley region were measured at JHU Tectonics, Metamorphic Petrology, and Orogenesis (TeMPO) Laboratory using a Teldyne-Cetac Analyte G2 193 nm Excimer laser ablation system, with a double volume HelEx II cell, coupled to an Agilent 8900 quadrupole LA-ICP-MS with no gas in the collision cell. Each analyzed fabric includes 15–59 measured spots. Baseline-subtraction and instrument drift corrections were made with the commercially available Iolite software, using the U–Pb Geochronology data reduction scheme and the NIST612 (JHU) or NIST614 (UCSB) glass standard as the primary reference material. Following initial data processing in Iolite, a mass-bias correction to the $^{238}U/^{206}Pb$ ratios of each analysis was made such that the lower-intercept date of the Tera-Wasserburg discordia for limestone secondary reference materials measured simultaneously with the unknowns were accurately reproduced. At JHU, reference material limestones WC-1[63] and Duff Brown Tank[64] were used. At UCSB, WC-1[63], ASH15[65], and an in-house secondary reference travertine Whitepine were used. The accuracy of carbonate U–Pb dates (probably no better than ~ ±3% at 95% C.I.) is currently limited by heterogeneity of available reference materials, such as WC-1 which exhibits ≥2.5% scatter about best fit discordia[63], as well as uncertainties surrounding differences in the ablation characteristics of different carbonate materials[63,66]. This limitation in method accuracy does not influence the results of this study, for which calculated date uncertainties (analytical and regression) are often quite large and possible error in calculated U–Pb dates

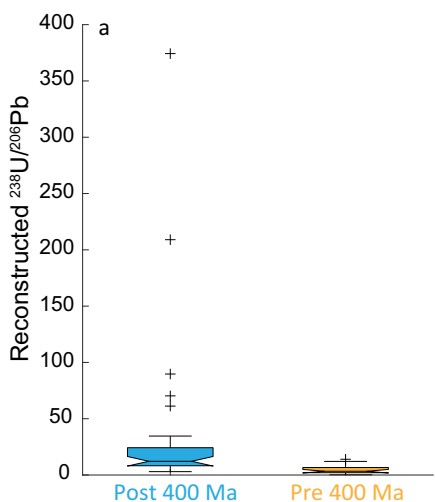
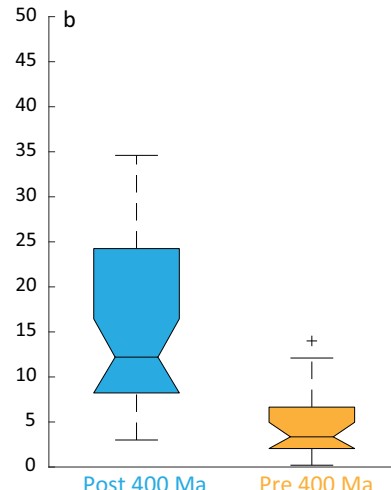

**Fig. 4 | boxplots summarizing reconstructed $^{238}$U/$^{206}$Pb values in post- and pre-400 Ma samples. a** full range of $^{238}$U/$^{206}$Pb values. **b** $^{238}$U/$^{206}$Pb values up to 50. Samples with stratigraphic ages <400 Ma average 34.9 and have median of 12.2 (center line) with 95% confidence interval of [16.4, 7.9] (notch). Samples with stratigraphic ages >400 Ma average 4.8 have median of 3.4 (center line) with 95% confidence interval of [5.0, 1.7] (notch). Mean values of post and pre 400 Ma samples are significantly different ($t$-test: $p = 2.8476e-05$). Box limits are upper and lower quartiles, whiskers are maximum and minimum extent of data and crosses are outliers. Source data are provided as a Source Data file.

on the order of a few million years are unimportant for the overall interpretations.

We consider these data together with published $^{207}$Pb/$^{206}$Pb and $^{235}$U/$^{238}$U ratios from the eastern Levant region ($n = 7$)[32], the northwest Sichuan Basin ($n = 5$)[34], the Arabian Platform[33] ($n = 11$) and the Wasatch Range, Utah ($n = 8$)[31]. For consistency, we calculated $^{238}$U/$^{206}$Pb and $^{207}$Pb/$^{206}$Pb ratios and (re)plotted all data in Tera-Wasserburg concordia plots to determine the U–Pb age of the samples and the initial $^{207}$Pb/$^{206}$Pb ratio[30]. We used an in-house Python based code to find the best fit line (discordia) through the data, and determine the U–Pb date, initial $^{207}$Pb/$^{206}$Pb, and their uncertainties[67]. For the latter, we used York least square method to calculate the uncertainty in the slope and intercept of isochrons that derives from analytical errors and natural scatter[68], and propagated these values to determine the uncertainty in age and intercept. Raw data of new measurements and Tera-Wasserburg plots of all analyzed data are available in Supplementary Dataset S3 and Fig. S1, respectively. Average $^{238}$U/$^{206}$Pb of the dolomite precursor mineral were calculated for all data ($n = 65$). Derivation for this calculation is given in Note S1, and the code is available online at[69]. We reported for all $^{238}$U/$^{206}$Pb values greater than zero ($n = 54$). Negative average $^{238}$U/$^{206}$Pb values emerge from samples having older than stratigraphic U–Pb ages or initial $^{207}$Pb/$^{206}$Pb ratios above the terrestrial evolution trend.

## Data availability
Source data are provided with this paper. Summary of measured and compiled U-Pb data is available in Supplementary Dataset S1. Details for samples collected by the authors from the Colorado Plateau and Death Valley are given in Supplementary Dataset S2. All measured raw data is available in Supplementary Dataset S3. Tera-Wasserburg plots for measured and compiled data are available in Fig. S1. Source data are provided with this paper.

## Code availability
Python code to process raw data to best fit discordia lines, slope (corresponding to date), intercepts (corresponding to initial $^{207}$Pb/$^{206}$Pb) and their uncertainties is available in Code Ocean archive: https://doi.org/10.24433/CO.0924321.v1[67]. Matlab code to calculate $^{238}$U/$^{206}$Pb of dolomite precursor minerals is available in Code Ocean archive: https://doi.org/10.24433/CO.8028801.v1[69].

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

## Acknowledgements
This work has been supported by Israel Science Foundation Grant 1010/20 to UR and Johns Hopkins University funding to EFS. UR thanks Ranjani Murali and Theodor M. Present for insightful discussions. We thank the U.S. National Park Service at Death Valley National Park for a sampling permit to LLN and EFS (#DEVA-2017-SCI-0006).

## Author contributions
M.B.I. and U.R. designed the research. R.M.H., L.L.N. and E.F.S. contributed to sample collection and context. R.M.H. and U.R. conducted laboratory analyses. U.R. and M.B.I. conducted data analysis, and interpretation and authored the manuscript. R.H.H. participated in writing the methods section of the manuscript. M.B.I., R.M.H., L.L.N., E.F.S., A.R.C.K.C. and U.R. contributed comments and input to the manuscript.

## Competing interests
The authors declare no competing interests.

## Additional information
**Supplementary information** The online version contains Supplementary Material available at https://doi.org/10.1038/s41467-024-46660-7.

