## [Peer Review File · Nature Communications]

Late Paleozoic Oxygenation of Marine Environments Supported by Dolomite U-Pb DatingREVIEWER COMMENTS

Reviewer #1 (Remarks to the Author):

See attached review, which I signed.

Reviewer #2 (Remarks to the Author):

Review of Late Paleozoic Oxygenation of Marine Environments Supported by Dolomite U-Pb Dating
In this paper, the authors present a dataset that captures U-Pb dates of dolomites of various ages and various precursor sediment ages, in order to reconstruct their initial $^{207}\text{Pb}/^{206}\text{Pb}$ ratios. The data suggest that the ages of the dolomite are younger than expected and the $^{207}\text{Pb}/^{206}\text{Pb}$ ratios are lower than expected, which is interpreted to reflect resetting of the U-Pb system coinciding with U decay, allowing for reconstruction of initial $^{238}\text{U}/^{206}\text{Pb}$ ratios of the rock. The study finds that $^{238}\text{U}/^{206}\text{Pb}$ ratios did not dramatically increase until the late Paleozoic, which they attribute to oxygenation of seawater, and ultimately providing evidence in support of the hypothesis that early animal life evolved within an oxygen-limited ocean until the late Paleozoic (a hotly debated topic). I find this study interesting and of broad interest to the scientific community—appropriate for a Nature Communications publication. However, I have some comments and suggestions that the authors might take into consideration prior to acceptance for publication. I will be upfront and admit that I am no expert on the U-Pb system and especially as it applies to carbonates (which appears to be a relatively newer but intriguing development for the carbonate discipline). Rather, my review focuses on the aspects of dolomitization, dolomite textures, and dolomite recrystallization—for which my background is rooted in. I hope the authors find my comments useful to help improve their manuscript, I would be happy to provide any further clarification.

Major Comments:

Line 74 the authors state "Using in-situ laser ablation inductively coupled plasma mass spectrometry, U and Pb isotope measurements were made on a variety of dolomite mineral fabrics, associated with depositions and early diagenetic environments (fabric-retentive cements) or with late diagenetic environments (fabric-destructive cements), while avoiding void-filling epigenetic dolomite (e.g., veins)."

However, my question is: Are they actually cements? Or are they replacement products? An important distinction, since in the former case they would NOT be inheriting precursor sediment geochemistry (they would be reflecting the fluids responsible for cementation), and would not accurately capture the interpretations that underpin the importance of this manuscript. However, in the latter case (i.e. the dolomites are replacive) there is potential to retain the precursor mineral proxy records. If the latter, you could replace these clarifiers with "(replacive and fabric-retentive)" and "(replacive and fabric-destructive)".

As an aside, fabric-destructive dolomite is not always indicative of "late diagenetic environments"-- numerous examples of fabric-destructive dolomite that has been interpreted to form in early diagenetic environments exist (planar-e sucrosic dolomite that does not retain the precursor fabric is extremely common).

Line 131 the authors state "Dolomite samples in our data compilation include fabric-retentive dolomites, which are though to be associated with early dolomitization^{43,44}."

I recommend replacing "which are thought" with "that have been interpreted". I suggest this change so the reader does not presume that you are suggesting that fabric retentive-dolomites are thought to be associated with early dolomitization. Rather, the fabric-retentive dolomites in your compilation have been studied by others who subsequently interpreted that they are associated with early dolomitization.

If you do mean the former (a generalization about fabric retentive-dolomites), I will refer you to

numerous high-temperature dolomitization experiments in which precursor fabrics were preserved-- which would somewhat refute the idea that early diagenetic environments (i.e., near-surface temperatures and conditions) are required for preservation (e.g., Bullen and Sibley 1984 [https://doi.org/10.1130/0091-7613\(1984\)12<655:DSAMR>2.0.CO;2](https://doi.org/10.1130/0091-7613(1984)12<655:DSAMR>2.0.CO;2) or Zempolich and Baker 1993 <https://doi.org/10.1306/D4267B86-2B26-11D7-8648000102C1865D>).

Lines 136-137 The authors state "Within this sample set, there are also indications of open-system isotope exchange of carbonate oxygen, and re-setting of clumped isotope compositions in deep burial environments³⁹."

When I first read this sentence, I was a bit confused because this is the first instance in which you are interpreting the early diagenetic fabric-retentive dolomites have been recrystallized in deep burial environments (which should apparently not be confused with fabric-destructive dolomites that also crystallized in deep burial environments), but you do not clearly state as such. I would recommend changing this sentence to something of the like: "Recrystallization of the fabric-retentive dolomites would be in line with studies included in our dataset, which indicate open-system isotope exchange of carbonate oxygen, and re-setting of clumped isotope compositions in deep burial environments ³⁹."

Lines 202-203 the authors state "Other dolomite textures, associated with precursor mineral deposition at the surface (ooids, peloids, and fossil fragments) or that are fabric-destructive (massive), record significantly higher values of reconstructed $^{238}\text{U}/^{206}\text{Pb}$ that are expected under more oxygenated depositions environments only after ~ 400 Ma (Fig. 3a)."

I think that you are here specifically trying to refer to non-microbial-associated dolomites, which you should explicitly say. Because what you say here "other dolomite textures associated with precursor mineral deposition at the surface, or that are fabric-destructive" means very little considering this describes nearly all textures with a broad stroke that is this sentence. I'm not sure whether you are trying to pivot here to non-microbial-associated dolomite, or to replacive dolomite, but you should make it clear rather than saying "other dolomite textures" and describing majority of the dolomite textures that exist.

Reviewer #3 (Remarks to the Author):

This manuscript by Ben-Israel and colleagues presents new U-Pb in dolomite data, which provides new insight into U-Pb geochronology in carbonates and the oxygenation history of the Earth. Although I am not an expert in these methods, the data appear analytically sound and the paper is well-written and informative.

My primary comment on the paper concerns its overall pitch towards the Earth's oxygenation record. Although this topic is near and dear to my heart, I found the implications for U-Pb dating in dolomites (for instance, being able to discern the time of alteration) much more interesting, and likely the most useful part of the study. I agree with the authors that their results do support late Paleozoic oxygenation, but looking at the Figure 3, there are already a large and diverse set of datasets that support this conclusion, while the results here are relatively widely spaced in time and space. So without trying to take anything away from the study, I found the implications for oxygenation to be interesting (and supportive of what I believe is the correct trajectory) but not that conclusive compared to the other available datasets. I also took from statements in the paper that the main thing that affects the U-Pb trend is the seawater U concentration, and if so, one could develop a much larger and better-coverage dataset simply by looking at [U] in carbonates. It is up to the authors, but my personal feeling was that the implications for dolomite geochronology were the more exciting aspect, and it might be better pitched as a geochronology paper.

No matter how the authors want to focus their paper, I think the questions of oxygenation should be bolstered by simple statistical analyses and data reporting regarding the primary hypothesis that $^{238}\text{U}/^{206}\text{Pb}$ ratios increased at 400 Ma. This should include reporting the average, median and 95% CI of the pre-400 and post-400 Ma interval, and the correct parametric or non-parametric statistical test (based on normality) of whether the younger samples are higher. Just eyeballing the data they look significantly and substantially larger but that needs to be demonstrated.

In the abstract and to some extent the introduction I think that, as written, the authors have set up a false dichotomy between Neoproterozoic oxygenation/animal evolution and later Paleozoic oxygenation. I believe there are two separate questions here: did an oxygen increase influence or 'trigger' the Cambrian radiation? And when did the ocean/atmosphere system become fully oxygenated to modern levels? The critical ecological thresholds that would have been important for the Cambrian radiation are likely in the 3-10% PAL O_2 range (discussed in (Sperling et al., 2022, 2015; not exhorting citation of these papers, rather explaining where my thoughts are published). A sustained change in atmospheric O_2 from say 4% to 10% PAL would likely be difficult to detect in most global datasets, especially when they are just presented in a 'shotgun blast' form like Fig. 3. And it would clearly not be the full oxygenation of the ocean-atmosphere system. But it would have had a dramatic effect on early animal ecosystems, including the advent of predators, and could have triggered the Cambrian radiation. So, I think it is very possible that a muted rise in oxygen influenced Cambrian evolution and that full oxygenation occurred in the Devonian. Changing the writing will not require much, but I do think it is important to get this distinction right.

I found Figure 2 to be very useful in my understanding and had a few suggestions that might help make it even clearer to people (like me) from outside this system. I would note in the first line of the figure caption that this is a 400 million year old sample that was altered at 200 million years ago. I would also drop the "~" on each of these, and just work the example such that it is 400 and 200, as the 'approximate' introduces unneeded confusion. On panel a I would also say that the deposition is at 400 Ma. Finally, I was a little confused about the "isochron from ~400 Ma after deposition" in panel c. Is that the 'ideal' isochron that would result if there was no alteration or homogenization? Could that be clarified?

I have some additional comments below, but otherwise I very much enjoyed reading this manuscript. Sincerely,

Erik Sperling
Stanford University

Line 15- Reference 7 and 10 are inappropriate here for this statement on oxygenation and Neoproterozoic-Cambrian animal evolution, as they are focused on the GOE. The authors may have meant (Planavsky et al., 2014) Science instead of Nat. Geosci, which would be more appropriate here.

Line 18 and also line 41- (Dahl et al., 2010; Wallace et al., 2017) should be cited here with respect to a later Paleozoic rise in atmospheric oxygen. Although they have been superseded in terms of the size of the datasets available, these two papers were really important in putting that idea out there.

Figure 1- Have the authors plotted the age offset as a function of absolute age? It doesn't really look like there is a trend, which is presumably interesting in and of itself for carbonate geochron.

Line 145—please very briefly give a little more geological detail. Presumably these are the Wasatch range samples, and you should say (Utah, USA) for those not familiar with the Alta stock.

Line 185- Modeling in (Johnson et al., 2014), which to my knowledge is the most recent on this topic, demonstrates that U is remarkably sensitive to atmospheric oxygen. These results suggest U oxidation

has been quantitative even farther back than the Neoproterozoic, to the GOE.

Line 190- (Lau et al., 2016) should also be cited here in addition to the Elrick paper, especially as there is excellent modeling of how U concentrations relates to global anoxia in the supplement to that paper.

Figure 3- Plotting the Berner GEOCARBSULF results is fine, but those are a little out of date. I would suggest the updated curves from (Mills et al., 2023) as more realistic.

Line 224- I am not too familiar with Pb redox cycling, could that be explained a little further?

Line 241- Lu et al. is cited twice with different numbers

Line 247-248: I think the possibility of minor or muted oxygenation is just as important, from a biological standpoint, if not more so, than 'short lived'

Line 289- calculated

References:

- Dahl, T.W., Hammarlund, E.U., Anbar, A.D., Bond, D.P.G., Gill, B.C., Gordon, G.W., Knoll, A.H., Nielsen, A.T., Schovsbo, N.H., Canfield, D.E., 2010. Devonian rise in atmospheric oxygen correlated to the radiation of terrestrial plants and large predatory fish. *Proceedings of the National Academy of Sciences, U.S.A.* 107, 17911–17915.
- Johnson, J.E., Gerpheide, A., Lamb, M.P., Fischer, W.W., 2014. O₂ constraints from Paleoproterozoic detrital pyrite and uraninite. *Geological Society of America Bulletin* B30949.1. <https://doi.org/10.1130/B30949.1>
- Lau, K.V., Maher, K., Altiner, D., Kelley, B.M., Kump, L.R., Lehmman, D.J., Silva-Tamayo, J.C., Weaver, K.L., Yu, M., Payne, J.L., 2016. Marine anoxia and delayed Earth system recovery after the end-Permian extinction. *PNAS* 113, 2360–2365. <https://doi.org/10.1073/pnas.1515080113>
- Mills, B.J.W., Krause, A.J., Jarvis, I., Cramer, B.D., 2023. Evolution of Atmospheric O₂ Through the Phanerozoic, Revisited. *Annual Review of Earth and Planetary Sciences* 51, 253–276. <https://doi.org/10.1146/annurev-earth-032320-095425>
- Planavsky, N.J., Reinhard, C.T., Wang, X., Thomson, D., McGoldrick, P., Rainbird, R.H., Johnson, T., Fischer, W.W., Lyons, T.W., 2014. Low mid-Proterozoic atmospheric oxygen levels and the delayed rise of animals. *Science* 346, 635–638. <https://doi.org/10.1126/science.1258410>
- Sperling, E.A., Boag, Thomas H., Duncan, M.I., Endriga, C.R., Marquez, J.A., Mills, D.B., Monarrez, P.M., Sclafani, J.A., Stockey, R.G., Payne, J.L., 2022. Breathless through time: oxygen and animals through Earth's history. *Biological Bulletin* 243, 184–206.
- Sperling, E.A., Knoll, A.H., Girguis, P.R., 2015. The ecological physiology of Earth's second oxygen revolution. *Annual Review of Ecology, Evolution, and Systematics* 46, 215–235. <https://doi.org/10.1146/annurev-ecolsys-110512-135808>
- Wallace, M.W., Hood, A., Shuster, A., Greig, A., J. Planavsky, N., Reed, C., 2017. Oxygenation history of the Neoproterozoic to early Phanerozoic and the rise of land plants. *Earth and Planetary Science Letters* 466, 12–19. <https://doi.org/10.1016/j.epsl.2017.02.046>

Reviewer #4 (Remarks to the Author):

I appreciate that the authors took data that might otherwise be considered a failure and tried to do something creative with it. I like that they took a systematic approach to looking at their own data and also published work from the literature. They have provided their code and others might use that to consider similar problems. They have done a nice job of tying this to other studies that seek to consider oxygenation of the atmosphere/ocean through time. They certainly have as much or more

data and are on no shakier ground than these other studies.

I do however have a number of reservations about the thought process that went into getting to the place where they think their calculated U/Pb ratio has something to do with the oxygenation of the ocean.

First, the assumption that good fabric preservation probably required early diagenetic stabilization is reasonable, but poor fabric preservation could be early or late, there is no way to know that. Second, there appears to be an underlying assumption that these were originally dolomite and have been recrystallized to a secondary dolomite. I don't see any justification for the original dolomite. I think it could have been any carbonate mineral. I believe the assessment of mineralogy and diagenesis needs to be carefully reworded to avoid making unsubstantiated statements.

Second, there is an assumption that the U is in the oxidized state. Based on a number of studies on UXANES it is not at all clear that this is the case. For sure dolomite is almost never a primary mineral forming directly from seawater. That means it is buried and altered in a different environment than the open ocean. We know from pore water studies across the ocean, that pore waters quickly become reducing. Reduced U appears to have a much higher K_d for carbonates than oxidized U.

Third, the U/Pb ratio is only part of the story. What are the concentrations? The paper seems to imply that redox has some control on Pb. This is not likely in a near surface environment. The U/Pb is not just a U concentration problem, it is a Pb concentration problem.

Finally, while it is reasonable to assume that the initial Pb isotope ratio of seawater has changed in a systematic way that follows the Pb growth curve, fluids that have interacted with other rocks along the path to dolomitization could be more radiogenic. Certainly the fluids don't have to be homogenizing the local carbonate (dolomite) as implied by the model.

Point-by-point reply to reviewers' comments (replies in green)

Reviewer #1 (Remarks to the Author):

The notion of long-delayed comprehensive oxygenation of the biosphere is relatively new and exciting. Research has moved that landmark event/interval from the GOE to the NOE and now well into the Paleozoic. It is remarkable to think that overall marine oxygen deficiencies persisted across an interval marked by the most significant steps in innovation and diversification of animal life. The model is gaining acceptance, although the evidence for that long protraction of oxygenation is still embryonic and demands more work and additional approaches. For example, relationships and decouplings between shallow and deep waters are critical and harder to assess.

The authors have taken this challenge on. The idea of looking at U-Pb relationships in carbonates for this purpose is absolutely novel, and, in a first-order sense, seems to line up with other approaches/arguments. Importantly, it does so in very independent ways that confirm but also broaden our understandings through mechanistic underpinnings that speak to global inventories of redox sensitive species rather being a proxy for local shallow or deep conditions. As such, I support the work very much and hope it will be published. But I have questions/comments:

- (1) The approach taken is complicated. The authors have worked hard to make the details accessible, but they will not be easy or even possible to follow for many readers outside the field. I very much encourage rereads and edits with an eye toward even greater clarity. For example, unless I missed it, we are never given the essential overarching detail that ^{238}U decays to stable ^{206}Pb . I know, basic stuff, but still useful to include. And this is just one example of where clarity could be massaged.

We went over the paper and made several modifications aiming to make it more accessible to non-expert readers. Edits include: 1) U nuclides decay to Pb daughters is now explained (lines 64-65); 2) Figure 2 was modified for clarity; 3) We added some details on location and geological context for non-experts (lines 154-155); 4) We added a geological time scale to figure 3; 5) we spell out 'billions of years' and 'millions of years' for Ga and Ma, respectively.

- (2) Those who follow this field will tell you that carbonate-hosted proxies in old rocks are the most vulnerable to alteration and corresponding push back from the community. This paper not only focuses on carbonate but specifically acknowledges and incorporates the effects/signatures of primary signal resetting in the arguments. These relationships are addressed through extensive conversations about the how, when, and the significance of age resetting and specifically ingrowth of ^{206}Pb through decay following resetting that scales with ^{238}U availability in the sample. With the added ability to look at the differences between stratigraphic ages and U-Pb dates, we can in theory back out initial U inventory (U/Pb for seawater specifically) in ways that speak directly to the global ocean redox state. I think I get all this (maybe not), and we are given many discussions about Pb exchange and potentially confounding factors tied to varying primary carbonate mineralogies, dolomite fabrics/generations, etc. But I am still missing, and probably this is my fault, why fundamentally the U inventory is also not reset during alteration, making it very hard to solve for the initial. What am I missing? We are told about resetting of the U/Pb system, but I am struggling to see how that overprint is resolved for U too. Too many variables? Much of the discussion focuses on Pb controls. Does the approach allow for resolvable loss and addition of both

isotopes? What about diagenesis (recrystallization) in pore waters selectively striped of U? Again, there are a lot of pieces to this story, and we are taken on a journey through multiple alternative hypothesis followed by arguments that challenge those. This is good. I suppose what makes me most excited about this paper is that while the details do make the arguments complicated and even suspicious, they do not make the first order U/Pb predictions in Fig. 3 unbelievable. In other words, I see the scope for mess, but I also see a first-order jump well into the Paleozoic that generally jibes with the other data types summarized in that figure. Again, it is this consistency that is tantalizing and elegant because of the very different approach taken for these rocks. Although I would like to hear more about what specifically this method brings to the table relative to other data types.

The U reservoir in our model is free to reset in open or closed system reactions during the alteration event in which age is reset. Considering the high mobility of this element – it is likely altered in the process. This is stated in lines 125-128: “In this process, U concentrations could also have been re-distributed and partly homogenized in either open or closed system conditions, but some U/Pb heterogeneity was maintained allowing U-Pb Discordia to be defined”. Importantly this does not interfere with the interpretation of initial 238/206 in the dolomite precursor, as this specific ‘memory’ is carried by two values in the dolomite: 1) the difference between the measured initial 207/206 and the expected initial at the time of deposition (Based on Stacey and Kremers, 1975 model); and 2) the time gap between depositional and U-Pb age. The U-Pb system is assumed to remain closed after modification. This assumption is supported by the fact that we obtain an isochron – that is, that lower 207/206 values are supported by higher 238/206 – meaning that U has decayed to lead and both parent and daughter remained in a closed system.

Regarding the possibility of U (and Pb) ratio altering in sediment/pore water profiles – this is certainly an option that is likely responsible to much of the scatter we see among samples from the same rock units. In the revised paper, we now acknowledge that U/Pb signal may be recorded in sediment under less-oxic conditions than the water column (lines 209-232, 246, 252-254). This issue has been mentioned in the previous version with respect to some samples. In the revised paper we now broaden the argument to include all samples – with the point that, regardless of the specific crystallization site (sediment or surface) suggested by dolomite texture, we observe high ²³⁸U/²⁰⁶Pb values only after 400 Ma – suggesting a rise in the levels of oxygen of marine surface and sediments at that time.

(3) At the end of the day, I think this is an argument about an increased U inventory in the ocean in the late Paleozoic. Fair enough. But I was very surprised by the absence of any mention of the Partin et al. (2013, EPSL) paper, which convincingly shows a U inventory increase recorded in black shales much earlier—in the latest Proterozoic. Something needs to be said about the apparent disagreement of these data types and what it might mean.

Good points, thanks you for directing us to this paper. Different patterns of redox-sensitive elements, including the step-wise increase during the Neoproterozoic observed by Partin et al. (2013) are briefly discussed in lines 255-270 – we made sure to include a reference to this paper in the revised version. One possible reason for the differences between the black shale and carbonate records is that these records capture different marine environments, which were possibly oxidized at different times. Specifically, Lu et al. (2018) suggested that shallow remineralization depth of organic material has kept marine shelves oxygen-poor well into the Phanerozoic, while deeper marine basins have been relatively oxygenated (Figure 2b in their paper). Another possibility is that rock records have been affected by post depositional alteration – e.g., recent oxidative weathering (Albut et al., 2018; Planavsky et al., 2020). These ideas were

implied in the previous version of the paper and are now made explicit on lines 259-262: **“These differences may reflect the complex structure of the early Paleozoic oxycline, in which shallow remineralization of organic material reduced continental shelves and left deeper basins relatively oxidized (Lu et al., 2018), or possibly different sensitivities of elements and the records that contain them to post-depositional alteration (Planavsky et al., 2020).”**

(4) Line 47, the authors seem content to put a cloud of doubt over a massive amount of past data by simply citing the Slotznick et al. challenge to a much early hypothesized whiff of oxygen. That challenge is not very substantial (see recent Anbar et al. rebuttal), and to suggest that this (the only cited paper) throws into doubt so much other work is just silly. In other words, the authors might work to more convincingly make a case for why their very complex approach applied to dolomite is better or at least needed—as in providing additional or different constraints, etc. Ironically, the most convincing part of their story may be that it agrees with other data types. (I noted the absence the Dahl et al. Mo isotope paper that many years ago set the stage for protracted oxygenation arguments.)

The possibility that redox proxies are altered by post depositional processes is certainly not limited to Slotznick et al. or the Archean “whiffs” of oxygen. Probably the most suitable example is the recent paper by Planavsky et al. (2020), in which post depositional alteration of redox signal, and specifically secondary oxidative weathering are considered (and demonstrated) to have affected Neoproterozoic and Early Paleozoic records. We are not saying that such alteration is necessarily the case, but that it should be considered as one way to explain the differences between records. In the introduction, we simply point out that this issue has been a major source for concern in the community. We agree with the reviewer that the conflicting schedules of Neoproterozoic-Paleozoic oxygenation warrant new approaches to this problem – and our approach is certainly a new one – but think that its greatest strength and source for justification is that it **“...is based on the observed initial $^{207}\text{Pb}/^{206}\text{Pb}$ ratio, which is relatively impervious to alteration after dolomite (re)crystallization, and specifically, to false-positive oxygenation signals (higher $^{238}\text{U}/^{206}\text{Pb}$) recorded as samples were approaching the surface during exhumation.”** We made sure to refer to the paper of Dahl et al. (2010).

(5) In my view, the authors way overstep the implications of their results. First, the U-Pb data show lots of scatter, wide error bars, and poor temporal coverage. There is no way the hypothesized late Paleozoic step, particularly as recorded in their data, can be used to infer that most of the major innovation in the history of animal life (the GOBE, infaunal revolution, predation and largeness, complexly layered ecologies, skeletonization, motility, etc.) happened under persistently very low oxygen levels in surface oceans. And even if they did, the U-Pb data say very little in this regard. The idea of persistently very low oxygen surface waters is hard to grasp and is out of phases with recent redox studies for the Ordovician coming out of the Florida State group, etc. (not cited in this paper).

We agree with the reviewer that our approach has several notable disadvantages (low temporal resolution, large error bars etc.). However, with these caveats, it can still constrain an oxidation schedule that is consistent with some higher-resolution records, and inconsistent with others – namely, Neoproterozoic step-wise major oxygenation of marine environments. Considering the main advantage of this approach (impervious to ‘false-positive’ oxygenation signals), this results lends credibility to the former, and thereby support hypotheses suggesting the persistent oxygenation of marine environments happened only after the Early Paleozoic (which is different from saying that anoxia was persistent before the Early Paleozoic). We revise the text to make this point clearer: **“...In this context, the fact that**

reconstructed $^{238}\text{U}/^{206}\text{Pb}$ ratios, which are resistant to the effects of ‘false positive oxygenation’, do not appear to record oxygenation signals during the Neoproterozoic and Cambrian within multiple samples from different formations and different paleoenvironments, at a first order, supports the model that persistent oxygenation of shallow and deep marine environments was delayed until the Devonian (Fig. 3).”

A bigger question is the relationship between the deep and shallow ocean that best fits all the observations, including the certain role of oxygen in driving early-mid Paleozoic evolution. The U-Pb data simply cannot speak to long-term persistence and prevalence of low oxygen in Paleozoic surface waters or its variation on shorter time scales. The required resolution for temporal, spatial, and persistence patterns is missing. Shorter-term variation could be huge part of the story but would go undetected in this approach. If I am not mistaken, the U-Pb data, like many other approaches, capture whole ocean relationships that can be controlled by evolution of the deep ocean with poorly understood relationships to the surface ocean where biological innovation took place.

We agree with the reviewer. Considering the limitations of the U-Pb and other records, the argument we now make is more nuanced than ‘major innovation in the evolution of animal life have occurred under very low oxygen levels’. As mentioned above, we do not argue for persistently low oxygen levels before 400 Ma; but rather for persistently high levels of oxygenation after 400 Ma. We made sure to note that, **“This scenario does not necessarily preclude the possibilities of minor (<10 % or the modern oxic ocean), local, and/or episodic (1-10 Ma) oxygenation events during the NOE, which did not register in the compiled $^{238}\text{U}/^{206}\text{Pb}$ and Ce/Ce* records shown here, but could have had critical ecological impacts vital for the Cambrian radiation (Sperling et al., 2015)”**. Animal life could have evolved in response to elevated oxygen levels in these episodes or environments, or possibly under minor rise in oxygen that remained ‘muted’ in some of the redox records.

We agree that the last sentence in the abstract of the previous version was over-simplifying our interpretations. We have changed this sentence to say: **“...This timeline is consistent with evolution-driven mechanisms for the oxygenation of late Paleozoic marine environments and with suggestions that Neoproterozoic and early Paleozoic animals thrived in oceans that overall and on long time scales were oxygen-limited compared to the modern ocean.”** We also revised the discussion to accommodate the range of possible scenarios that explain the lack of oxygenation signal prior to 400 Ma: **“(our data, considered together with consistent records) ... It is also consistent with previous suggestions that several major evolutionary and ecological steps in the diversification of animal life took place within a largely oxygen-limited ocean, in which O_2 concentrations were sufficient to overcome functional and ecological thresholds, but insufficient to register in a series of redox proxies, possibly since they were minor (<10%) or existed temporary and/or locally relative to the modern oxic ocean.”**

(6) The error bars in Fig. 1 are huge. And while I see the general relationships suggested by the authors, such as the tendency for data to fall below the 1:1 line with dates significantly younger than the strat. age, the plot suggests that the results are at best a first-order indication of very general relationships. These errors/ranges may not be a problem, but they deserve a comment or two.

The uncertainties on several U-Pb ages are indeed significant. This is typical of dolomite samples that naturally have low uranium content and an overall small range of $^{238}\text{U}/^{206}\text{Pb}$. This is not a problem in the analysis or interpretation in which the signal is well above the statistical noise (Fig. 3a, and new data repository figure S1). As suggested, we commented on the large uncertainties in this figure caption. ***“Large uncertainties in U-Pb dates are common to carbonate samples that typically suffer from low ranges of $^{238}\text{U}/^{206}\text{Pb}$.”***

(7) I would include geologic ages and well as absolute ages on the X-axis of Fig. 3.

Adding the geological timescale is a great idea – done. Each measured dolomite point has both stratigraphic and U-Pb age that are separated by 10-100's of Ma. I'm afraid that Including both (say if we define the x axis as stratigraphic or U-Pb age and connect the dots with a line) will make this panel and the figure too busy. Mostly, we're afraid that it will cause confusion, since the reconstructed $^{238}\text{U}/^{206}\text{Pb}$ values are attributed to the dolomite precursor mineral forming at or near the stratigraphic age – not the time of alteration.

Again, because of the cleverness of the approach and its complementarity to other data types in critical discussions about delayed biospheric oxygenation, I support eventual publication. But I would hope for a dialed-back pitch about the implications of the results with more emphasis and balance placed instead on the promise of the approach and the questions still remaining. In truth, if it weren't for the general agree with the conceptual models previously informed by the other data types in Fig. 3, we might be saying that altered dolomite with reset U-Pb chemistry is not a good target. But I do applaud the creativity and boldness of the approach and don't disagree with the overarching conclusions.

Tim Lyons UC Riverside

Reviewer #2 (Remarks to the Author):

Review of Late Paleozoic Oxygenation of Marine Environments Supported by Dolomite U-Pb Dating
In this paper, the authors present a dataset that captures U-Pb dates of dolomites of various ages and various precursor sediment ages, in order to reconstruct their initial $^{207}\text{Pb}/^{206}\text{Pb}$ ratios. The data suggest that the ages of the dolomite are younger than expected and the $^{207}\text{Pb}/^{206}\text{Pb}$ ratios are lower than expected, which is interpreted to reflect resetting of the U-Pb system coinciding with U decay, allowing for reconstruction of initial $^{238}\text{U}/^{206}\text{Pb}$ ratios of the rock. The study finds that $^{238}\text{U}/^{206}\text{Pb}$ ratios did not dramatically increase until the late Paleozoic, which they attribute to oxygenation of seawater, and ultimately providing evidence in support of the hypothesis that early animal life evolved within an oxygen-limited ocean until the late Paleozoic (a hotly debated topic).

I find this study interesting and of broad interest to the scientific community—appropriate for a Nature Communications publication. However, I have some comments and suggestions that the authors might take into consideration prior to acceptance for publication. I will be upfront and admit that I am no expert on the U-Pb system and especially as it applies to carbonates (which appears to be a relatively newer but intriguing development for the carbonate discipline). Rather, my review focuses on the aspects of dolomitization, dolomite textures, and dolomite recrystallization—for which my background is rooted in. I hope the authors find my comments useful to help improve their manuscript, I would be happy to provide any further clarification.

Major Comments:

Line 74 the authors state “Using in-situ laser ablation inductively coupled plasma mass spectrometry, U and Pb isotope measurements were made on a variety of dolomite mineral fabrics, associated with depositions and early diagenetic environments (fabric-retentive cements) or with late diagenetic environments (fabric-destructive cements), while avoiding void-filling epigenetic dolomite (e.g., veins).” However, my question is: Are they actually cements? Or are they replacement products? An important distinction, since in the former case they would NOT be inheriting precursor sediment geochemistry (they would be reflecting the fluids responsible for cementation), and would not accurately capture the interpretations that underpin the importance of this manuscript. However, in the latter case (i.e. the dolomites are replacive) there is potential to retain the precursor mineral proxy records. If the latter, you could replace these clarifiers with "(replacive and fabric-retentive)" and "(replacive and fabric-destructive)".

As an aside, fabric-destructive dolomite is not always indicative of "late diagenetic environments"-- numerous examples of fabric-destructive dolomite that has been interpreted to form in early diagenetic environments exist (planar-e sucrosic dolomite that does not retain the precursor fabric is extremely common).

Good point – The term ‘cement’ was used incorrectly in the previous version. Following the nomenclature in Sibely and Gregg (1987) all analyzed fabrics are now defined as replaced matrix or allochems components. We made sure to correct that. We tweaked the sentence to account for the possibility of fabric destructive dolomites have been forming at in early or later diagenetic environments. It now reads:

“Using in-situ laser ablation inductively coupled plasma mass spectrometry, U and Pb isotope measurements were made on a variety of replacive dolomite mineral fabrics, associated with depositional and early diagenetic environments (fabric-retentive) or possibly with late diagenetic environments (fabric-destructive), while avoiding void-filling epigenetic dolomite (e.g., veins).”

Having said that, as mentioned above our analyses brings into account the possibility of dolomite precursory minerals forming from pore water at the subsurface – as they might do in a cement. It is possible that samples in our dataset have been affected by shallow diagenetic redox to various extents. This is what can explain the scatter we see within and among dolomites associated with different formation environments and processes (massive vs. surface vs. microbial textures). But, this scatter is minor when compared with the signal of post 400 Ma oxygenation, that is apparent within each of these groups.

Line 131 the authors state “Dolomite samples in our data compilation include fabric-retentive dolomites, which are thought to be associated with early dolomitization^{43,44}.”

I recommend replacing "which are thought" with "that have been interpreted". I suggest this change so the reader does not presume that you are suggesting that fabric retentive-dolomites are thought to be associated with early dolomitization. Rather, the fabric-retentive dolomites in your compilation have been studied by others who subsequently interpreted that they are associated with early dolomitization.

If you do mean the former (a generalization about fabric retentive-dolomites), I will refer you to numerous high-temperature dolomitization experiments in which precursor fabrics were preserved--

which would somewhat refute the idea that early diagenetic environments (i.e., near-surface temperatures and conditions) are required for preservation (e.g., Bullen and Sibley 1984 [https://doi.org/10.1130/0091-7613\(1984\)12<655:DSAMR<2.0.CO;2](https://doi.org/10.1130/0091-7613(1984)12<655:DSAMR<2.0.CO;2) or Zempolich and Baker 1993 <https://doi.org/10.1306/D4267B86-2B26-11D7-8648000102C1865D>).

Good catch – sentence now reads: “**Dolomite samples in our data compilation include fabric-retentive dolomites that have been interpreted as products of early dolomitization (Corsetti et al., 2006; Nelson et al., 2021)**”.

Lines 136-137 The authors state “Within this sample set, there are also indications of open-system isotope exchange of carbonate oxygen, and re-setting of clumped isotope compositions in deep burial environments³⁹.”

When I first read this sentence, I was a bit confused because this is the first instance in which you are interpreting the early diagenetic fabric-retentive dolomites have been recrystallized in deep burial environments (which should apparently not be confused with fabric-destructive dolomites that also crystallized in deep burial environments), but you do not clearly state as such. I would recommend changing this sentence to something of the like: "Recrystallization of the fabric-retentive dolomites would be in line with studies included in our dataset, which indicate open-system isotope exchange of carbonate oxygen, and re-setting of clumped isotope compositions in deep burial environments ³⁹."

Done – with a minor tweak to say that isotope evidence for late crystallization come from both fabric retentive and destructive dolomites.

Lines 202-203 the authors state “Other dolomite textures, associated with precursor mineral deposition at the surface (ooids, peloids, and fossil fragments) or that are fabric-destructive (massive), record significantly higher values of reconstructed ²³⁸U/²⁰⁶Pb that are expected under more oxygenated depositions environments only after ~400 Ma (Fig. 3a).”

I think that you are here specifically trying to refer to non-microbial-associated dolomites, which you should explicitly say. Because what you say here "other dolomite textures associated with precursor mineral deposition at the surface, or that are fabric-destructive" means very little considering this describes nearly all textures with a broad stroke that is this sentence. I'm not sure whether you are trying to pivot here to non-microbial-associated dolomite, or to replacive dolomite, but you should make it clear rather than saying "other dolomite textures" and describing majority of the dolomite textures that exist.

‘Non microbial-associated dolomites’ is indeed a better way of putting it. We changed the text as suggested.

Reviewer #3 (Remarks to the Author):

This manuscript by Ben-Israel and colleagues presents new U-Pb in dolomite data, which provides new insight into U-Pb geochronology in carbonates and the oxygenation history of the Earth. Although I am not an expert in these methods, the data appear analytically sound and the paper is well-written and

informative.

My primary comment on the paper concerns its overall pitch towards the Earth's oxygenation record. Although this topic is near and dear to my heart, I found the implications for U-Pb dating in dolomites (for instance, being able to discern the time of alteration) much more interesting, and likely the most useful part of the study. I agree with the authors that their results do support late Paleozoic oxygenation, but looking at the Figure 3, there are already a large and diverse set of datasets that support this conclusion, while the results here are relatively widely spaced in time and space. So without trying to take anything away from the study, I found the implications for oxygenation to be interesting (and supportive of what I believe is the correct trajectory) but not that conclusive compared to the other available datasets. I also took from statements in the paper that the main thing that affects the U-Pb trend is the seawater U concentration, and if so, one could develop a much larger and better-coverage dataset simply by looking at [U] in carbonates. It is up to the authors, but my personal feeling was that the implications for dolomite geochronology were the more exciting aspect, and it might be better pitched as a geochronology paper.

See response to point #2 and 4 of reviewer #1. The ability to determine the timing of alteration of the U-Pb system in carbonates has already been demonstrated before (Gasparrini et al., 2023; Mangenot et al., 2018). In the revised version, we now highlight the strength of our approach (being resistant to 'false positive' oxygenation signals') and its contribution to our understanding of other proxy records and the overall history of marine oxygenation (lines 16-18, 209-212, 262-267); we also added a short paragraph at the very end of the paper – in which suggest that the approach developed here is likely to work just as well in other authigenic mineral records (lines 171-176). Previous studies have looked at the concentration of [U] in carbonates directly, and report an increase around the NOE (See compilation in Chen et al., 2021). The different patterns may reflect some difference in formation environments between the set of dolomite samples we studied and the limestone included in the compilation of Chen et al. (2021) – or possibly post depositional alteration of U in the limestone samples. To have a better idea, we will need to measure U-Pb dates and [U] concentrations on the same samples. Alteration of U in carbonates is a likely scenario. For example, diagenetically altered carbonates have very different [U] and [Pb] concentrations and overall lower U/Pb compared with primary biogenic carbonates (Roberts et al., 2020).

No matter how the authors want to focus their paper, I think the questions of oxygenation should be bolstered by simple statistical analyses and data reporting regarding the primary hypothesis that $^{238}\text{U}/^{206}\text{Pb}$ ratios increased at 400 Ma. This should include reporting the average, median and 95% CI of the pre-400 and post-400 Ma interval, and the correct parametric or non-parametric statistical test (based on normality) of whether the younger samples are higher. Just eyeballing the data they look significantly and substantially larger but that needs to be demonstrated.

Post and pre 400 Ma samples have significantly different mean and median values. To demonstrate this, we added a box plot as a supplementary material figure S1– and included relevant statistics parameters and test results in the caption (see below)

Figure S1 – boxplot summarizing reconstructed $^{238}\text{U}/^{206}\text{Pb}$ values in Post- and Pre-400 Ma samples. A) full range of values. B) focusing on values up to 50. Samples with stratigraphic ages <400 Ma average 34.9 and have median of 12.2 with 95% confidence interval of [16.4, 7.9]. Samples with stratigraphic ages >400 Ma average 4.8 have median of 3.4 with 95% confidence interval of [5.0, 1.7]. Mean values of post and pre 400 Ma samples are significantly different (t-test: $p=2.8e-05$).

I think that, as written, the authors have set up a false dichotomy between Neoproterozoic oxygenation/animal evolution and later Paleozoic oxygenation. I believe there are two separate questions here: did an oxygen increase influence or ‘trigger’ the Cambrian radiation? And when did the ocean/atmosphere system become fully oxygenated to modern levels? The critical ecological thresholds that would have been important for the Cambrian radiation are likely in the 3-10% PAL O₂ range (discussed in (Sperling et al., 2022, 2015; not exhorting citation of these papers, rather explaining where my thoughts are published). A sustained change in atmospheric O₂ from say 4% to 10% PAL would likely be difficult to detect in most global datasets, especially when they are just presented in a ‘shotgun blast’ form like Fig. 3. And it would clearly not be the full oxygenation of the ocean-atmosphere system. But it would have had a dramatic effect on early animal ecosystems, including the advent of predators, and could have triggered the Cambrian radiation. So, I think it is very possible that a muted rise in oxygen influenced Cambrian evolution and that fully oxygenation occurred in the Devonian. Changing the writing will not require much, but I do think it is important to get this distinction right.

We agree, and revised the introduction (line 38-44) and discussion (lines 267-271, 278-283) to make the distinction clear.

I found Figure 2 to be very useful in my understanding and had a few suggestions that might help make it even clearer to people (like me) from outside this system. I would note in the first line of the figure caption that this is a 400 million year old sample that was altered at 200 million years ago. I would also drop the “~” on each of these, and just work the example such that it is 400 and 200, as the ‘approximate’ introduces unneeded confusion. On panel a I would also say that the deposition is at 400 Ma. Finally, I was a little confused about the “isochron from ~400 Ma after deposition” in panel c. Is that the ‘ideal’ isochron that would result if there was no alteration or homogenization? Could that be

clarified?

We corrected figure 2 and its caption following the suggestions above.

I have some additional comments below, but otherwise I very much enjoyed reading this manuscript. Sincerely,

Erik Sperling
Stanford University

Line 15- Reference 7 and 10 are inappropriate here for this statement on oxygenation and Neoproterozoic-Cambrian animal evolution, as they are focused on the GOE.

The authors may have meant (Planavsky et al., 2014) Science instead of Nat. Geosci, which would be more appropriate here.

Good catch – we indeed meant the science paper. Catling and Clair’s paper removed as suggested.

Line 18 and also line 41- (Dahl et al., 2010; Wallace et al., 2017) should be cited here with respect to a later Paleozoic rise in atmospheric oxygen. Although they have been superseded in terms of the size of the datasets available, these two papers were really important in putting that idea out there.

Done.

Figure 1- Have the authors plotted the age offset as a function of absolute age? It doesn’t really look like there is a trend, which is presumably interesting in and of itself for carbonate geochron.

Good idea – It didn’t occur to us earlier – below is the plot, of course that with older stratigraphic age larger offsets are possible (the limit is marked by the 1:1 line). Otherwise there is no apparent trend. It would take many more data points and extending the range of dated dolomites to explore this quantitatively.

Line 145—please very briefly give a little more geological detail. Presumably these are the Wasatch range samples, and you should say (Utah, USA) for those not familiar with the Alta stock.

We added 'In dolomite samples' and "(Utah, USA)" as suggested.

Line 185- Modeling in (Johnson et al., 2014), which to my knowledge is the most recent on this topic, demonstrates that U is remarkably sensitive to atmospheric oxygen. These results suggest U oxidation has been quantitative even farther back than the Neoproterozoic, to the GOE.

Good point, we revised the ending of this sentence so that: **"...It is not likely that such a change is the result of variation in the supply of U and Pb to the ocean, as oxidative continental weathering of these elements has been dominant since the early Proterozoic (Holland, 1984; Johnson et al., 2014)"**

Line 190- (Lau et al., 2016) should also be cited here in addition to the Elrick paper, especially as there is excellent modeling of how U concentrations relates to global anoxia in the supplement to that paper. Reference to Lau et al. (2016) was added as suggested.

Figure 3- Plotting the Berner GEOCARBSULF results is fine, but those are a little out of date. I would suggest the updated curves from (Mills et al., 2023) as more realistic.

We actually switched from GEOCARBSULF to Mill's (2023) trend just before submission and forgot to update the caption. It is updated now. Thank you for noticing.

Line 224- I am not too familiar with Pb redox cycling, could that be explained a little further?

In oxygenated environments, dissolved Pb is scavenged by Fe-Oxyhydroxides and removed from solution – for example see paper by (Kalnejais et al., 2015). We now make a note of it in lines 203-207.

Line 241- Lu et al. is cited twice with different numbers

Good catch – we got rid of one of the doppelgangers.

Line 247-248: I think the possibility of minor or muted oxygenation is just as important, from a biological standpoint, if not more so, than 'short lived'

We agree. We rephrased the sentence to: **"This scenario does not necessarily preclude the possibilities of minor (<10 % or the modern oxic ocean), local, and/or episodic (1-10 Ma) oxygenation events during the NOE, which did not register in the compiled $^{238}\text{U}/^{206}\text{Pb}$ and Ce/Ce* records shown here, but could have had critical ecological impacts vital for the Cambrian radiation (Sperling et al., 2015)".** We believe that "muted" is covered by the part of the sentence saying "which did not register in..."

Line 289- calculated

Corrected as suggested by the reviewer.

References:

Dahl, T.W., Hammarlund, E.U., Anbar, A.D., Bond, D.P.G., Gill, B.C., Gordon, G.W., Knoll, A.H., Nielsen, A.T., Schovsbo, N.H., Canfield, D.E., 2010. Devonian rise in atmospheric oxygen correlated to the

radiation of terrestrial plants and large predatory fish. *Proceedings of the National Academy of Sciences, U.S.A.* 107, 17911–17915.

Johnson, J.E., Gerpheide, A., Lamb, M.P., Fischer, W.W., 2014. O₂ constraints from Paleoproterozoic detrital pyrite and uraninite. *Geological Society of America Bulletin* B30949.1. <https://doi.org/10.1130/B30949.1>

Lau, K.V., Maher, K., Altiner, D., Kelley, B.M., Kump, L.R., Lehrmann, D.J., Silva-Tamayo, J.C., Weaver, K.L., Yu, M., Payne, J.L., 2016. Marine anoxia and delayed Earth system recovery after the end-Permian extinction. *PNAS* 113, 2360–2365. <https://doi.org/10.1073/pnas.1515080113>

Mills, B.J.W., Krause, A.J., Jarvis, I., Cramer, B.D., 2023. Evolution of Atmospheric O₂ Through the Phanerozoic, Revisited. *Annual Review of Earth and Planetary Sciences* 51, 253–276. <https://doi.org/10.1146/annurev-earth-032320-095425>

Planavsky, N.J., Reinhard, C.T., Wang, X., Thomson, D., McGoldrick, P., Rainbird, R.H., Johnson, T., Fischer, W.W., Lyons, T.W., 2014. Low mid-Proterozoic atmospheric oxygen levels and the delayed rise of animals. *Science* 346, 635–638. <https://doi.org/10.1126/science.1258410>

Sperling, E.A., Boag, Thomas H., Duncan, M.I., Endriga, C.R., Marquez, J.A., Mills, D.B., Monarrez, P.M., Sclafani, J.A., Stockey, R.G., Payne, J.L., 2022. Breathless through time: oxygen and animals through Earth's history. *Biological Bulletin* 243, 184–206.

Sperling, E.A., Knoll, A.H., Girguis, P.R., 2015. The ecological physiology of Earth's second oxygen revolution. *Annual Review of Ecology, Evolution, and Systematics* 46, 215–235. <https://doi.org/10.1146/annurev-ecolsys-110512-135808>

Wallace, M.W., Hood, A., Shuster, A., Greig, A., J. Planavsky, N., Reed, C., 2017. Oxygenation history of the Neoproterozoic to early Phanerozoic and the rise of land plants. *Earth and Planetary Science Letters* 466, 12–19. <https://doi.org/10.1016/j.epsl.2017.02.046>

Reviewer #4 (Remarks to the Author):

I appreciate that the authors took data that might otherwise be considered a failure and tried to do something creative with it. I like that they took a systematic approach to looking at their own data and also published work from the literature. They have provided their code and others might use that to consider similar problems. They have done a nice job of tying this to other studies that seek to consider oxygenation of the atmosphere/ocean through time. They certainly have as much or more data and are on no shakier ground than these other studies.

I do however have a number of reservations about the thought process that went into getting to the place where they think their calculated U/Pb ratio has something to do with the oxygenation of the ocean.

First, the assumption that good fabric preservation probably required early diagenetic stabilization is reasonable, but poor fabric preservation could be early or late, there is no way to know that.

Good point. We now acknowledge the possibility of fabric destructive dolomites have been forming at in early or later diagenetic environments (line 79).

Second, there appears to be an underlying assumption that these were originally dolomite and have been recrystallized to a secondary dolomite. I don't see any justification for the original dolomite. I think

it could have been any carbonate mineral. I believe the assessment of mineralogy and diagenesis needs to be carefully reworded to avoid making unsubstantiated statements.

This is the opposite of the assumption we make – the samples are considered to modify through dolomitization (e.g., from Ca-carbonate mineral) or dolomite recrystallization (from protodolomite) (Line 109-112). Later in the paper (lines 138-146), dolomite crystallization is preferred over dolomitization to explain the seeming contradiction between early dolomitization (suggested in previous publications) and large gaps between U-Pb and depositional ages. Our model will work equally well if the U-Pb system is reset during deep dolomitization rather than recrystallization. Further research and larger data-sets are required to identify specific alteration pathways and associate them with geochemical signal-alteration trends. In the revised discussion, we made sure to broaden the argument regarding the possibility of crystallization of dolomite-precursor minerals in sediments under conditions less-oxic than overlying seawater (lines 212-223, 246, 252-254).

Second, there is an assumption that the U is in the oxidized state. Based on a number of studies on UXANES it is not at all clear that this is the case. For sure dolomite is almost never a primary mineral forming directly from seawater. That means it is buried and altered in a different environment than the open ocean. We know from pore water studies across the ocean, that pore waters quickly become reducing. Reduced U appears to have a much higher K_d for carbonates than oxidized U.

Indeed, recent experiments and observations, support that U^{4+} is preferably incorporated to calcite minerals in hydrothermal and near surface environments (Gabitov et al., 2021; Gulbranson et al., 2022). However, being particle reactive U^{4+} tends to be quickly immobilized in environments <100 °C (Gabitov et al., 2021) – limiting the supply of U to the crystallization site in oxygen-limited environments. In the context of our study, we argue that the supply of U^{6+} to the crystallization site (in which it may be reduced to U^{4+} before incorporation to calcite) impose a first order control over the observed variation in reconstructed $^{238}U/^{206}Pb$ values - rather than preferential uptake of U^{4+} to calcite crystal lattice. This is for three reasons:

First, the concentrations of U in carbonates is reported to decrease during global marine anoxia events (Elrick et al., 2017; Lau et al., 2016) – These events have been associated with the reduction of U^{6+} to U^{4+} in seawater – based on observed decrease in the $\delta^{238}U$ values (Lau et al., 2016). If the concentration of uranium in carbonate was set by the U partition factor, then we would expect to see more uranium in calcite at the same intervals. The fact that we don't, suggests that the supply of dissolved U^{6+} from seawater to the carbonate crystallization site is controlling U concentration in carbonate minerals (lines 207-208).

Second, dolomite samples in our dataset preserve primary (laminar, microbial) textures which have been associated with crystallization of dolomite precursor minerals under reducing pore water environments. If the U/Pb ratio in the precursor mineral was controlled by preferential partition of U to calcite, then we would expect to see the highest $^{238}U/^{206}Pb$ ratios in these samples (as U^{4+} is preferentially incorporated) when compared with dolomite fabrics associated with surface deposits (fossils, ooids etc.). However, we observe the exact opposite, as such fabrics appear to record relatively low $^{238}U/^{206}Pb$ in the Jurassic (fig. 3A in the manuscript) which is consistent with the supply of U and possibly Pb from seawater being the determining factor (lines 216-226).

And third, the overall temporal trend we observe is of increasing $^{238}\text{U}/^{206}\text{Pb}$ during the Phanerozoic. If reconstructed $^{238}\text{U}/^{206}\text{Pb}$ ratio in the dolomite precursor mineral was dictated by preferential incorporation of U^{4+} under reducing conditions, then we would expect the reverse trend.

Third, the U/Pb ratio is only part of the story. What are the concentrations? The paper seems to imply that redox has some control on Pb. This is not likely in a near surface environment. The U/Pb is not just a U concentration problem, it is a Pb concentration problem.

Our approach can only reconstruct the $^{238}\text{U}/^{206}\text{Pb}$ ratio (which is proportional to the U/Pb ratio) – not absolute U or Pb concentrations. We now briefly refer to Pb redox sensitivity (see reply to reviewer #3). And yes, as mentioned above we agree that the U/Pb record reflect carbonate crystallization environments which can be (and are even likely to be) pore water. While the incorporation of Pb to carbonate mineral lattices is yet another under-constrained question, the fact that U/Pb ratio in dolomites associated (by texture) with reducing environments register relatively low reconstructed $^{238}\text{U}/^{206}\text{Pb}$ values, and that the $^{238}\text{U}/^{206}\text{Pb}$ signal rise, rather than falls between the Proterozoic and Phanerozoic confirms that this ratio is redox sensitive. Throughout the paper we broaden the argument regarding formation under reducing pore water environments to explicitly include the possibility that the oxygenation in shallow diagenetic environments may have driven this signal.

Finally, while it is reasonable to assume that the initial Pb isotope ratio of seawater has changed in a systematic way that follows the Pb growth curve, fluids that have interacted with other rocks along the path to dolomitization could be more radiogenic. Certainly the fluids don't have to be homogenizing the local carbonate (dolomite) as implied by the model.

When considering single result, we cannot rule out this possibility. But since the terrestrial evolution trend reflects an average of the crust, we would expect to see values both above and below the trend if that was the case. The fact that the initial values are always offset towards lower $^{207}\text{Pb}/^{206}\text{Pb}$ suggests that the source naturally has higher U/Pb compared with the terrestrial average – e.g., it is likely to be siliciclastic or igneous which dominate the upper crust. We briefly refer to this source as 'river or deep seated groundwater' in lines (118-119).

REVIEWERS' COMMENTS

Reviewer #1 (Remarks to the Author):

I have carefully read through the response note and looked at the associated edits in the paper. There are four sets of thorough and constructive reviews, which address a wide diversity of comments and potential improvements. The authors have treated each with care, and I believe a better paper has resulted. I could quibble over some details, but there is no point to that. This paper presents a clever and novel approach that brings new light to important questions. There is enough here now for readers to process and place the results into their own views of the uncertainties and their overall thinking about Earth's protracted oxygenation and the implications for the history of life. This contribution dovetails nicely into that conversation. I am happy to sign off on this version.

Reviewer #2 (Remarks to the Author):

I commend the authors on adequately considering and addressing the comments I made on the previous version, and I have no further comments to make. I support this publication pending the decisions of the reviewers and their responses.

Brooks Ryan

Reviewer #3 (Remarks to the Author):

The authors have made the changes I suggested or explained why they did not. I appreciate the new Fig. S1, but to my reading at least, this is the paper, and it shouldn't be buried in the supplement!
Sincerely,
Erik Sperling

Reviewer #4 (Remarks to the Author):

I have read the revised version of this manuscript and I feel that the authors have done an adequate job of addressing my original questions. I think this is an interesting dataset and idea that should be published and is worthy of publication in Nature Communications.

Reviewer #4 (Remarks on code availability):

I have logged onto the private connection to the codes and found them easy to use.